# Tensile and Surface Wettability Properties of the Solvent Cast Cellulose Fatty Acid Ester Films

**DOI:** 10.3390/polym15122677

**Published:** 2023-06-14

**Authors:** Heikko Kallakas, Tanuj Kattamanchi, Catherine Kilumets, Elvira Tarasova, Illia Krasnou, Natalja Savest, Iman Ahmadian, Jaan Kers, Andres Krumme

**Affiliations:** 1Laboratory of Wood Technology, Department of Materials and Environmental Technology, School of Engineering, Tallinn University of Technology, Ehitajate tee 5, 19086 Tallinn, Estonia; 2Laboratory of Biopolymer Technology, Department of Materials and Environmental Technology, School of Engineering, Tallinn University of Technology, Ehitajate tee 5, 19086 Tallinn, Estonia

**Keywords:** cellulose, cellulose ester, solvent casting, films, tensile properties, wettability

## Abstract

Thermoplastic cellulose esters are promising materials for bioplastic packaging. For that usage, it is important to understand their mechanical and surface wettability properties. In this study, a series of cellulose esters are prepared, such as laurate, myristate, palmitate, and stearate. The aim of the study is to investigate the tensile and surface wettability properties of the synthesized cellulose fatty acid esters to understand their suitability as a bioplastic packaging material. Cellulose fatty acid esters are first synthesized from microcrystalline cellulose (MCC), then dissolved in pyridine solution, and after the solvent cast into thin films. The cellulose fatty acid ester acylation process is characterized by the FTIR method. Cellulose esters hydrophobicity is evaluated with contact angle measurements. The mechanical properties of the films are tested with the tensile test. For all the synthesized films, FTIR provides clear evidence of acylation by showing the presence of characteristic peaks. Films’ mechanical properties are comparable to those of generally used plastics such as LDPE and HDPE. Furthermore, it appears that with an increase in the side-chain length, the water barrier properties showed improvement. These results show that they could potentially be suitable materials for films and packaging materials.

## 1. Introduction

In recent years the demand for bio-derived resources because of overriding environmental contamination and diminishing fossil reserves worldwide has been brought to attention [1]. At present, polyethylene (PE) has extensive applications for packaging and a low price, which makes it the most broadly used thermoplastic. It is mostly obtained from petrochemistry. Polylactic acid (PLA), a biopolymer that has the potential to replace fossil-based materials, is becoming increasingly popular. However, its drawbacks are brittleness, low thermal stability, and a narrow processing window [1].

Biopolymers such as cellulose, starch, and chitin produce over 170 × 10^9^ tons of biomass, of which cellulose is 35–50%. This concludes that cellulose is the most abundant biopolymer. It is possible to chemically modify it, and the synthesis of cellulose esters allows for obtaining the needed strength and biodegradability and enables the use of cellulose for various applications. However, cellulose’s hydrophilicity, dense fibrous structure, and insolubility in traditional solvents are restricting its full potential for usage [1,2,3,4].

Cellulose esters are the most common derivatives. Many esters are commercialized, e.g., cellulose nitrate (CN), cellulose acetate (CA), cellulose acetate propionate (CAP), and cellulose acetate butyrate (CAB). These materials have a short chain length of <C6. Cellulose acetate is used in a variety of industries, including medicine, textiles, and cigarette filters. To improve thermoplastic properties, long-chain esterification can be used. They are synthesized from longer side-chain fatty acids (≥C6) [1,3,4]. Longer fatty acid esters include a side-chain length from C6 to C18. Because of their high price and complicated production process, they are not widely used commercially. However, there is a lot of potential for their use as bioplastics. They have higher strength, better water barrier properties, and a wider temperature processing window before they go into thermal decomposition territory [5]. It is important to recognize whether the solution is homogeneous or heterogeneous. The heterogeneous solution is more common since it is easier to produce and it is not necessary to dissolve cellulose completely. It is mostly used for esters, ethers, and silyl ethers. However, it is hard to control the degree of substitution (DS), which could lead to creating more waste [6,7]. Therefore, the homogeneous solution has better control over the structure and has successfully modified it to produce thermoformable materials [3,8,9]. Virtanen reported that stepwise esterification is useful for internally plasticised cellulose ester derivative preparation that is suitable for extrusion moulding and melt spinning with good mechanical properties. Their synthesized mixed cellulose ester hexanoate acetate showed plasticised polymer behaviour without the external plasticiser [10]. A more recent study investigated cellulose mixed esters production via a homogeneous reactive twin-screw extruder catalyzed by the ionic liquid. They managed to dissolve cellulose within minutes, resulting in the continuous production of cellulose acetate propionate [11]. To reduce petroleum-based chemicals, the Kusuma research group investigated methods to synthesise fully biobased cellulose esters by the oxidative esterification process. They used α,β-unsaturated aldehydes in an ionic liquid as solvent and catalyst. As a result, they reported having a fully substituted ester (DS = 3) with excellent atom economy without any catalysts or activators [12]. When it comes to plasticizers, they are most often used for cellulose acetates due to the high glass transition temperature (Tg) and narrow melt processing, which can be managed with the help of given additives. I.e., it has been reported that cellulose diacetate with triacetin can lower the Tg to 50 °C [13]. However, cellulose octanoate (C8) and palmitate (C16) are melted and processable without any additions [14]. The fatty acids in the long-chain cellulose ester already function as an internal plasticizer, reducing the need for any additional plasticizers [5].

Previous research has shown ductile behaviour as the side chain length and DS values increase [3,15,16]. Generally, the ranges are as follows: elastic modulus I, 150 to 350 MPa; tensile strength (σ), 8 to 20 MPa; and tensile strain (ε), 5 to 65% [15]. The elastic modulus E values are lowered with a longer side chain, making them more ductile. The given result is coherent since longer chains restrict materials’’ movement. Additionally, the DS increases tensile strength and strain level until a certain point. However, a newer source reports no major correlation between DS and its mechanical properties in fatty acid cellulose esters (FACEs) [17].

One of the important characteristics of the packaging materials is their hydrophobicity. Cellulose esters have shown good hydrophobic properties with contact angles over 89°. Additionally, the values are reported to be similar to those of commercially used polyethylene (i.e., LDPE, 91°) [15]. However, it is dependent on the ester’s composition. A higher degree of substitution and longer chain length increase the contact angle [15,17,18]. When discussing the matter of the influence of the degree of substitution even further, the reason for the increase in contact angle value is the influence of aliphatic chains. According to Wen [16], the esterification of MCC with a hydrophobic aliphatic chain results in a decrease in hydrophilic properties, especially if the hydrophilic groups are completely substituted.

Solvent casting is a fast and robust method to obtain results with a limited amount of material. Previous studies have focused a lot on the different solvent systems used to dissolve cellulose. Research has also shown that solvent-casted cellulose films can be affected by different process conditions such as drying conditions (wetting and barrier properties, optical properties), the surface roughness of the substrate (morphology and barrier properties), substrate hydrophobicity (transparency), and open-air or vacuum drying (crystallinity) [17,18,19]. However, the mechanical and surface properties (wettability and roughness) of the solvent-casted cellulose esters obtained by homogeneous transesterification with [mTBNH][OAc] ionic liquid have not been thoroughly studied. In this research, the aim is to investigate the tensile and surface hydrophobicity properties of the solvent-casted films of synthesized cellulose esters. The cellulose esters obtained were characterized by means of Fourier-transform infrared spectroscopy (FTIR), contact angle, and tensile test.

## 2. Materials and Methods

### 2.1. Preparation of Cellulose Fatty Acid Esters

The commercial cellulose diacetate was obtained from the electrospinning company Esfil Tehno AS, located in Sillamäe, Estonia. The longer side-chain cellulose fatty acid esters are synthesized from microcrystalline cellulose (MCC), the conditions of which are presented in Table 1, and the products are shown in Table 2. It was obtained from Carl Roth Gmbh & Co. Kg, Karlsruhe, Germany (CAS no. 9004-34-6) and has a density of 1.5 g/cm^3^, a pH value of 5–7, and a fibre length of 0.02–0.1 mm. Molar mass (MM) of pure cellulose was determined at 25 °C from the intrinsic viscosity [η] of cellulose solution in cupriethylenediamine hydroxide, CuEn, according to a standard procedure ASTM D1795-13. The MM was then calculated by the Mark-Houwink equation with parameters K = 1.01 × 10^−4^ dL/g and a = 0.9 [20]. The obtained MM was 163,000 g/mol. The MCC was used for the transesterification process with vinyl esters in ionic liquid (IL), i.e., 5-methyl-1,5,7-triaza-bicyclo-[4.3.0]non-6-enium acetate, [mTBNH][OAc], which was synthesized by Liuotin Group Oy, Taramäentie, Finland. The viscosity of MCC in IL was reduced using dimethyl sulfoxide (DMSO) as a co-solvent. Vinyl esters with a purity >98% were purchased from Tokyo Chemical Industry Co. (Tokyo, Japan). DMSO was purchased from Fisher Chemical, Waltham, MA, USA (CAS no. 67-68-5) and pyridine with a purity of ≥99.5% was acquired by Sigma-Aldrich, St. Louis, MO, USA (CAS no. 7291-22-7). Cellulose stearate (C18) synthesis differs from others (see Table 1) by having the conditions of 1:1 pyridine:IL at 80 °C for 3 h, the molar ratio is 1:6.

#### 2.1.1. Cellulose Dissolution and Esterification Procedure

MCC was dried under vacuum at 105 °C for 24 h before use. Additionally, 3.5 g of MCC was dissolved in 100 g of [mTBNH][OAc] and stirred at 60 °C for several hours until the cellulose was completely dissolved to yield a 3.5 wt% solution. To decrease the viscosity, which simplifies the processing, a co-solvent DMSO was added in a ratio of 1:1 to IL. The designed amount of the respective vinyl ester (3 eq./AGU) was added to the cellulose solution in a chemical reactor equipped with a mechanical stirrer and nitrogen flow, and then the reaction was performed at the conditions shown in Table 1. Cellulose stearate showed very low values of DS (0.6), and as a result, it was insoluble in any tested organic solvents. Therefore, an excess of vinyl stearate (5 eq./AGU) was used to drive the reaction to the right, increasing the production of ester. When the reaction was completed, the obtained cellulose esters were precipitated into 500 mL of water mixed with ethanol (for laurate and myristate) or pure ethanol (for palmitate and stearate). To remove solvent and vinyl ester residuals, the product was washed several times in 100–200 mL of ethanol, then acetone, and finally hexane. After that, the product was dried under a vacuum at 55 °C overnight.

#### 2.1.2. Determination of Degree of Substitution (DS)

DS values were determined with ^1^H NMR (hydrogen-1 nuclear magnetic resonance) and ^13^C NMR (carbon nuclear magnetic resonance) methods. The ^1^H NMR and ^13^C NMR spectra of the cellulose fatty acid esters were acquired on an Agilent Technologies DD2 500 MHz spectrometer equipped with 5 mm broadband inverse (^1^H) or broadband observe (^13^C spectra) probes. In addition, a 15 min temperature equilibration delay was allowed between sample insertion and NMR acquisition at 40 °C (cellulose palmitate and stearate in CDCl_3_) or 80 °C (cellulose laurate in DMSO_-d6_ and cellulose myristate and laurate in Py_-d5_) sample temperatures. Typically, for ^1^H spectra, 32 scans with a 25 s relaxation delay were acquired, and for ^13^C, 20,000–45,000 scans with a 2.5 s recycle delay were acquired to achieve the desired signal-to-noise ratio.

The chemical structures of the cellulose esters were confirmed by ^1^H NMR and ^13^C NMR. In the ^1^H NMR spectrum, the proton signals from approximately 6.00 to 3.50 ppm are assigned to protons of AGU in cellulose. The signals at 2.32–1.93, 1.88–1.45, and 1.27 ppm are associated with the methylene protons. The signals at 0.98–0.79 ppm are attributed to the methyl protons.

In the ^13^C NMR spectrum of cellulose esters, the signals from 34.78 to 23.34 are assigned to carbons of the aliphatic fatty acid chain, and 14.60 ppm is assigned to the carbon of the ending methyl group of the aliphatic side chain. The AGU carbons C-1, C-1′, C-4, C-2,3,5, C-6, and C-6′ give signals at 104.98, 102.33, 81.50, 76.91–74.35, 64.32, and 62.68 ppm, respectively. The signals from 173.89 to 170.98 ppm correspond to the carbonyl carbon at C-7, which provides direct confirmation of the successful attachment of the long-chain fatty acid chain to the cellulose backbone.

The DS of cellulose laurate, -myristate, -palmitate, and -stearate was calculated with equation 1 from the ^1^H NMR spectrum by taking an integral of the area of terminal methyl groups (ICH3) and AGU signals (IAGU) based on the reported method [19]. The calculated DS results are presented in Table 2.
(1)$$\mathrm{DS}\;=10\cdot \frac{I_{\left({\mathrm{CH}}_{3}\right)}}{\left(3\cdot I_{\mathrm{AGU}}+I_{\left({\mathrm{CH}}_{3}\right)}\right)}$$

### 2.2. Film Casting

Cellulose esters were dissolved in pyridine, and the obtained solutions shear rate ranged from 0.001 to 100 s^−1^ (Anton Paar PhysicaMCR501 rheometer, Ostfildern, Germany). Solutions were then solvent casted on laminated glass plates (250 mm × 150 mm) using the film casting knife (BYK-Gardner GmbH, Geretsried, Germany) with a blade width of 100 mm. With the given knife, the casted thickness of 200 μm was set to achieve the final film thickness of 70–100 μm. Except for cellulose diacetate, in which a casting thickness of 100 μm was set due to a higher cellulose concentration. Subsequently, the films were dried for 24 h in the open air under a fume at a room temperature of 20 °C. The films were removed using a phase immersion method in water after the solutions had been set on the glass plates. Additionally, the pH of the film’s surface was determined with the flat electrode using the Mettler Toledo SevenCompact pH meter S210 (Columbus, OH, USA) to check the acidic levels of each synthesized film. All measurements were conducted at a room temperature of 23 °C and an air humidity of 40%. Table 2 gives an overview of the solvent-casted films.

**Table 2 polymers-15-02677-t002:** Overview of solvent-casted films.

Nr	Cellulose Ester Full Name	Ester	Cellulose Concentration (%)	DS	pH
1	Cellulose acetate—commercial (C2)	Acetate_COM	15	2.48	4.67
2	Cellulose laurate (C12)	Laurate	7	1.5	5.50
3	Cellulose myristate (C14)	Myristate	7	1.7	4.86
4	Cellulose palmitate (C16)	Palmitate	7.1	0.8	4.51
5	Cellulose stearate (C18) *	Stearate	5.8	0.8	4.83

*—different synthesis process in the case of C18.

### 2.3. Characterization

The FTIR spectroscopy analysis was carried out on native cellulose and synthesized cellulose esters to evaluate the acylation process. Measurements were performed on an Interspectrum FTIR spectrometer (model Interspec 200-X, Tõravere, Estonia) with the KBr disc method between 500 and 4000 cm^−1^. Four scans were taken from each sample with a resolution of 8 cm^−1^ in absorbance mode. For comparison, all spectra were adjusted to the same baseline.

The surface hydrophobicity of the prepared films was estimated by measuring parameters such as equilibrium contact angle. The sessile drop method was used to measure contact angles with the device DataPhysics OCA 20 and the SCA 20 software (Riverside, CA, USA). All measurements were performed on the top side of the films with distilled water as a liquid agent to create a drop on the surface. The total observation time of the contact angle changing was 40 s starting from the moment when the water drop came into contact with the measured surface. An average of each point of measurement (0.2, 2, 5, 10, 20, 30, 40 s) was taken. A total of six measurements were taken for each variable, and the material average was taken from these six measurements. Before the measurement, the specimens were fixed with clamps on the measuring plate to achieve a smooth surface. All measurements of the contact angle were performed at room temperature and a relative humidity of 40%. 

The tensile test was performed according to ISO 527-3 [21]. The standards’ specimen type 1B (gauge dimensions of 50 mm × 10 mm) was used in an altered 1:2 ratio. The tensile properties were evaluated using a ZwickRoell Z050 (Ulm, Germany) universal testing machine with a load cell of 500 N. In the surrounding conditions of 23 °C and 50% relative humidity (RH), elastic modulus, tensile strength, and strain at break were determined. The grip distance and testing speed were 50 mm and 1 mm/min, respectively. For each variable, up to 6 specimens were tested. For the data analysis, a one-way ANOVA was subjected between all the groups with an alpha of 0.05. When the *p*-value was significant, the Bonferroni correction with the new alpha was utilized by dividing the alpha by the given comparisons (*t*-tests). The new alpha was set at 0.005 (10 comparisons).

## 3. Results

### 3.1. FTIR Characterisation

Figure 1 and Figure 2 present the comparative FTIR spectra of native microcrystalline cellulose and the synthesized cellulose esters. The synthesized cellulose fatty acid esters provide clear evidence of acylation with characteristic peaks. The intensive absorption bands at 2918 cm^−1^, 2849 cm^−1^, and 1467 cm^−1^ identify the presence of fatty acid long chains and correspond to anti-symmetric C−H stretching, symmetric C−H stretching, and C−H bending vibrations of the cellulose stearate, palmitate, myristate, and laurate [22]. In the case of the short chain length acetate (C2), these peaks are present with fewer intensities, and all the synthesized cellulose esters showed the three important distinct ester bands at 1738 cm^−1^ (C=O), 1225 cm^−1^ (−C−O− stretching), and 716 cm^−1^ (CH_2_ rocking, peculiar for at least four linearly connected −CH_2_− groups) [23]. These bands demonstrate successful acylation and correspond to ester carbonyl groups [15,17,24,25]. Furthermore, a significant decrease in the specific band for hydroxyl groups in the range of 3325–3480 cm^−1^ (OH stretching) is observed for all the synthesized fatty acid esters compared to the native microcrystalline cellulose.

### 3.2. Surface Wettability

In general, cellulose esters should be more hydrophobic than cellulose due to acetylation [26]. However, it has been shown that acetate solution cast film is not completely hydrophobic, and the contact angle is usually lower than 90° [27]. Hydrophobicity is the desired property in the food packaging industry as it gives resistance against liquids and water and ensures a longer shelf life of food products [28]. In order to assess the hydrophobicity of the different cellulose ester films, the surface wettability with contact angle measurement was determined. Figure 3 shows the average contact angle of different cellulose esters between 0.2 and 40 s. The contact angles increased with the length of the fatty chain, where C12 and C18 had values of 84.94° ± 0.23° and 104.01° ± 4.04°, respectively. C16 and C18 appear hydrophobic at angles higher than 90°. Previous studies have noted, with the increase in chain length and degree of substitution, that the surface of the cellulose esters becomes more hydrophobic [15,16,18]. In our study of the solvent-casted films, the data showed that with the increase of aliphatic side chains, the films got more hydrophobic. Additionally, the decrease in −OH groups leads to hydrophobicity, as shown by Wen [16]. This trend also corresponds with our research data. ANOVA analysis showed that the cellulose acetate contact angle is significantly lower (*p* < 0.05) compared to fatty acid cellulose esters (C14, C16, C18), which correlates with the observation of chain length increase. Thus, this research shows that all the synthesized cellulose fatty acid esters have the desired hydrophobic properties (contact angle >90°) for the use of food packaging materials.

### 3.3. Tensile Properties

Tensile properties of the casted films were observed to evaluate the effect of side chain length on cellulose fatty acid esters strength, elongation, and elastic properties. From our study results (Figure 4), it is shown that all cellulose ester casted films show ductile material characteristics, thus having a low resistance to deformation but showing both elastic and plastic deformation. A longer side chain, on the other hand, enhances stiffness. The stiffest ester in such circumstances is cellulose palmitate. On the stress-strain curve (Figure 4), it is seen that no strain softening occurs, i.e., a decrease in strength with an increase in strain. This implies that the deformation is uniform up to the point of failure. All the studied tensile behaviours of cellulose esters (Figure 4) correspond to previous research about cellulose laurate mechanical properties [29]. The stress-strain curves in Figure 4 show the elongation data (strain at break, e) of 40–85% for the C12–C18 samples (e = 40% for C16 and 85% for C12). This shows that the elongation of the cellulose fatty acid esters decreases with the sidechain length increase, which corresponds with the previously reported research data [15,17].

Figure 5 shows the average values of tensile strength and elastic modulus. Elastic modulus and strength increase with the chain length in fatty acid-cellulose ester-casted films. This outcome is contrary to that of Crépy and Willberg [15,18], who both showed that the modulus decreased with the increase of fatty acid side chain length. However, the DS values of their research were different from this study. Nevertheless, since the results are also dependent on the degree of substitution (DS), the samples with a lower DS show a higher elastic modulus, which corresponds with the previous studies [3,15]. Our data shows that cellulose palmitate with a low DS (0.81–0.83) has the highest elastic modulus (544.85 ± 11.78 MPa) compared to other synthesized cellulose fatty acid esters with greater DS values. As a result of the palmitate’s low DS, hydrogen bonding could be present due to a large number of free hydroxyl groups [3], and the material acts more similarly to commercial cellulose diacetate (in the elastic modulus aspect).

When compared to the commercially available cellulose acetate, the long sidechains of cellulose esters give a significant decrease in elastic modulus (~70–80%), tensile strength (~60–70%), and an increase in strain at break (~700–1200%). This data indicates that the fatty acids in the cellulose esters could already function as an internal plasticizer. According to Crépy and Wen [15,16], when there are more aliphatic chains in the structure, the material shows more ductile characteristics. Based on our research data, laurate showed −83%, −65%, and +1170% differences in elastic modulus, strength, and strain, respectively, compared to acetate.

Our research data show some similarities to those of commercial polymers (low-density polyethylene and high-density polyethylene). According to Crépy [15], LDPE has similar elastic modulus (102–240 MPa) and strength (7–16 MPa) ranges that correspond with our solvent-casted films. From the same study, HDPE has a lower elongation (50–900%) than LDPE (100–800%), which is more consistent with our casted film data. With its increased strength and elastic modulus, cellulose palmitate resembles HDPE more than LDPE. On the other hand, the most LDPE-resembling ester is cellulose laurate, with its highest elongation (85%).

## 4. Conclusions

New cellulose esters (laurate, myristate, palmitate, and stearate) were successfully synthesized in ionic liquid and solvent casted into films. Their surface and mechanical properties were evaluated and compared with those of commercial cellulose diacetate. A lower degree of substitution (<1) results in an increase in stiffness and strength (palmitate (C16)), which is more like the highly substituted cellulose acetate. However, with the effect of long aliphatic chains, the material obtains a higher strain than commercially used cellulose acetate, which is satisfactory for packaging material. All the synthesised cellulose fatty acid esters showed the desired hydrophobic properties for the food packaging materials. With the increase in side-chain length, the water barrier properties improved. Cellulose palmitate (C16) and cellulose stearate (C18) casted films showed the highest hydrophobic properties. When compared to commercial cellulose acetate (C2), the synthesised cellulose esters had increased elongation and hydrophobicity. Cellulose laurate (C12) and cellulose palmitate (C16) are most similar to LDPE and HDPE, respectively. According to our research data, it is promising that the fatty acid cellulose esters could be utilized for future bioplastic applications.

## Figures and Tables

**Figure 1 polymers-15-02677-f001:**
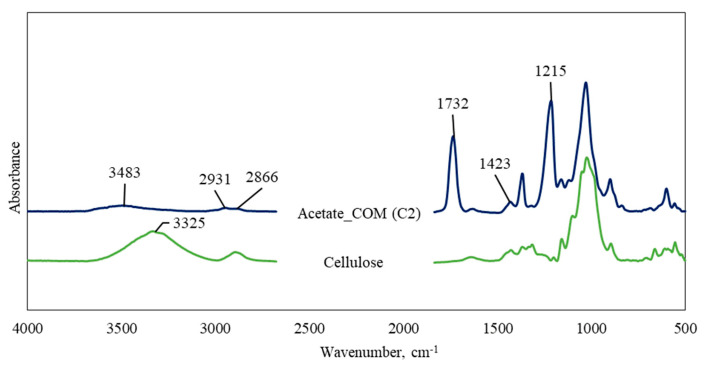
FT-IR spectra of native cellulose and commercial cellulose diacetate esters.

**Figure 2 polymers-15-02677-f002:**
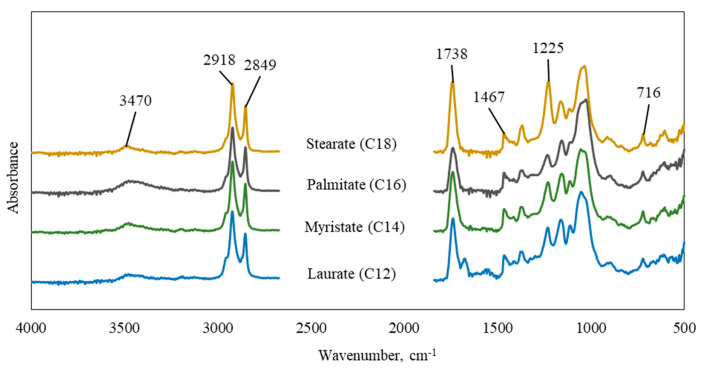
FT-IR spectra of synthesized cellulose esters.

**Figure 3 polymers-15-02677-f003:**
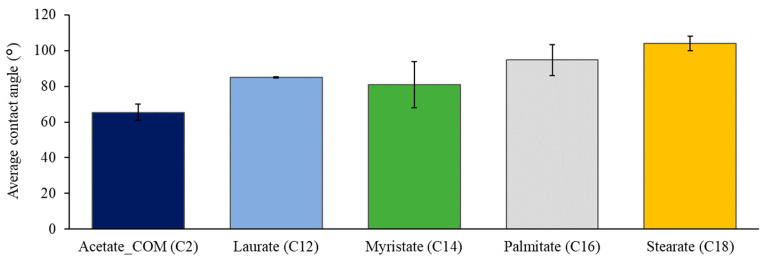
Average contact angles of cellulose esters.

**Figure 4 polymers-15-02677-f004:**
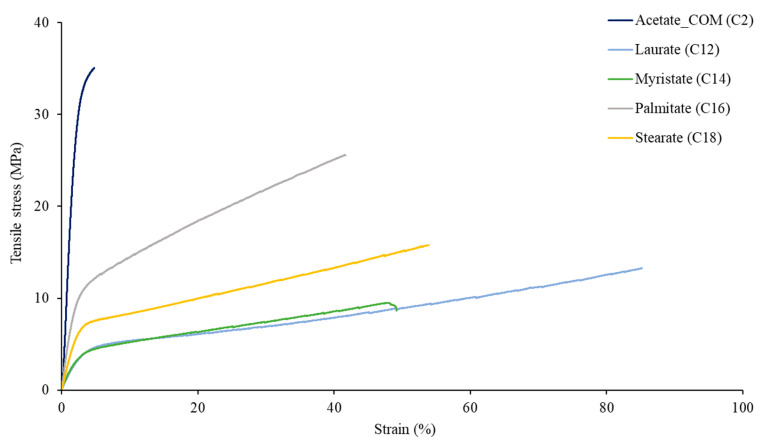
Stress-strain curves of cellulose esters.

**Figure 5 polymers-15-02677-f005:**
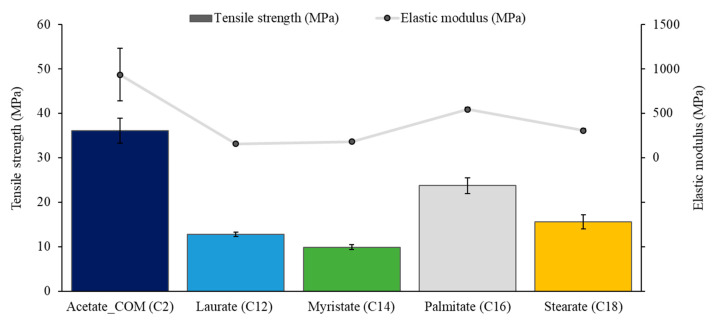
Average tensile strength and elastic modulus of cellulose esters.

**Table 1 polymers-15-02677-t001:** Cellulose esters synthesis reaction conditions (except C18).

Parameter	Condition
MCC content (%)	3.5
Solvent	1:1 DMSO:IL
Molar ratio of AGU:Vinyl ester	01:03
Temperature (°C)	70
Reaction time (h)	2.5

## Data Availability

The data presented in this study are available on request from the corresponding author. The data is not publicly available due to the confidentiality of the running project.

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
