# Peer review of "Tensile and Surface Wettability Properties of the Solvent Cast Cellulose Fatty Acid Ester Films"

_polymers, 2023, doi:10.3390/polym15122677_

Round 1
Reviewer 1 Report
In this manuscript, series of cellulose fatty acid esters were prepared using microcrystalline cellulose as a raw material and [mTBNH][OAc] ionic liquid as a solvent. The tensile and surface wettability properties of solvent casted cellulose ester films were investigated using contact angle measurements and tensile tests.
Fatty acid cellulose esters and their properties have been studied a lot in recent years and similar results have been published many times before. Also cellulose transesterification reaction in ionic liquid media have been earlier studied and published. Therefore, in my opinion the results of this manuscript are not new. Based on quality of the present work, I feel that this manuscript is not recommended for publication in Polymers.
In addition, this manuscript can already be found on the Research Square -portal (https://doi.org/10.21203/rs.3.rs-2191830/v1, submitted there in Oct 2022). I am not familiar with this portal, but it seems strange that this manuscript has been visible on the internet for six months.
Author Response
Response to Reviewer 1 Comments
Point 1: Fatty acid cellulose esters and their properties have been studied a lot in recent years and similar results have been published many times before. Also cellulose transesterification reaction in ionic liquid media have been earlier studied and published. Therefore, in my opinion the results of this manuscript are not new. Based on quality of the present work, I feel that this manuscript is not recommended for publication in Polymers.
Response 1: The product we obtained is the cellulose ester that is not intrinsically different from any cellulose esters obtained by other methods, but authors offer a novel sustainable strategy to get these CEs. In industry to obtain CEs such solvents as DMAc/LiCl, NMNO and others are used, which are old-fashioned and harmful and tend to be replaced with modern and greener alternatives. The usefulness and novelty of the current manuscript lies exactly in finding the alternatives to the old generation solvents for cellulose. The modern generation of solvents for cellulose is ionic liquids (ILs). EMIMAc is one of the most frequently employed ILs for dissolution and processing of cellulose. However, in addition to thermal degradation, (imidazolium) ILs can undergo side reactions with particular derivatization reagents that may lead to unexpected products. (https://doi.org/10.1007/s10570-018-2198-0). In our work we study applicability of [mTBNH][OAC] for cellulose esterification as far as it was found to be excellent in cellulose dissolution and fibre spinning, for example [https://doi.org/10.1016/j.fluid.2022.113414]. Also, it has significantly better stability for recycling procedures. It is not harmful, durable and recyclable. All those make this ionic liquid perspective for industrial application. In any case to be used in industrial upscaling novel IL should be comprehensively studied. Our study provides a useful knowledge about advantages and some drawbacks of [mTBN][OAC] and the possibility to use it for esterification of cellulose and production of long chain fatty acid cellulose esters. In addition, we are investigating these synthesized cellulose esters properties with tensile and surface wettability properties.
Point 2: In addition, this manuscript can already be found on the Research Square -portal (https://doi.org/10.21203/rs.3.rs-2191830/v1, submitted there in Oct 2022). I am not familiar with this portal, but it seems strange that this manuscript has been visible on the internet for six months.
Response 2: Polymers (MDPI) does permit existing preprints
Polymers accepts submissions that have previously been made available as preprints provided that they have not undergone peer review. A preprint is a draft version of a paper made available online before submission to a journal.
Polymers | Instructions for Authors (mdpi.com)
Reviewer 2 Report
In the manuscript "Tensile and surface wettability properties of the cellulose fatty acid esters" the authors presented novel cellulose esters and several aspects of comparison and characterization of those esters. The manuscript is very interesting, the hypotheses are adequate, and this article can be published after revisions and after addressing certain comments and ambiguities.
My main comments are:
1. Authors should include the number of measurements for contact angle and to precise if reported values of angles are changing or not during 40 s of measurement. Is standard deviation derived from values of contact angle taken in a range of 1-40 s or it is average of several measurements performed on certain number of samples? Please clarify
2. It would be interesting to add and correlate the crystallinity of produced films, especially related to mechanical properties.
3. Literature should be expanded to include the most relevant references, especially related to the production of cellulose esters and films from cellulose esters.
4. Discussion should be expanded to include other reports related to the properties of cellulose esters films.
5 In the title, authors should include a more specified description of the study, and to include that it is related to ester films.
Author Response
Response to Reviewer 2 Comments
Point 1: Authors should include the number of measurements for contact angle and to precise if reported values of angles are changing or not during 40 s of measurement. Is standard deviation derived from values of contact angle taken in a range of 1-40 s or it is average of several measurements performed on certain number of samples? Please clarify
Response 1: A total of six repetitions were done for each variable. Each measurement point (0.2, 2, 5, 10, 20, 30, 40 s) was averaged, and the material overall average and standard deviation was determined from these calculated values. The number of measurements was added to materials and methods.
Point 2: It would be interesting to add and correlate the crystallinity of produced films, especially related to mechanical properties.
Response 2: Thank you for the suggestion. We would definitely like to add this information next time. Unfortunately, we did not measure the crystallinity of the samples and it is hard to go back to this at the moment.
Point 3: Literature should be expanded to include the most relevant references, especially related to the production of cellulose esters and films from cellulose esters.
Response 3: Literature references about related to the production of cellulose esters and films from cellulose esters were added in the Introduction.
Point 4: Discussion should be expanded to include other reports related to the properties of cellulose esters films.
Response 4: Discussion part was modified and some new references were added.
Point 5: In the title, authors should include a more specified description of the study, and to include that it is related to ester films.
Response 5: More specific title was added: “Tensile and surface wettability properties of the solvent cast cellulose fatty acid ester films”
Reviewer 3 Report
In this study, authors studied the mechanical properties and hydrophobicity of the cellulose derivatives as the side chain length changes. The paper is acceptable for publication after major revisions of several issues. The comments and grammatical problems are listed in the pdf files near the corresponding highlighted places. One main issue is the FTIR spectrum of acetate_COM. No signals are observed in the range of 2800-4000 cm-1. Authors need to remeasure the samples with more scans and compared their results with that in references. The reason why no signals for -OH, CH3- groups and CH2- groups should be presented to confirm authors's conclusion.

English language needs to be checked thoroughly or seek for help from a native English speaker to present clearer meaning.
Author Response
Response to Reviewer 3 Comments
Point 1: In this study, authors studied the mechanical properties and hydrophobicity of the cellulose derivatives as the side chain length changes. The paper is acceptable for publication after major revisions of several issues. The comments and grammatical problems are listed in the pdf files near the corresponding highlighted places. One main issue is the FTIR spectrum of acetate_COM. No signals are observed in the range of 2800-4000 cm-1. Authors need to remeasure the samples with more scans and compared their results with that in references. The reason why no signals for -OH, CH3- groups and CH2- groups should be presented to confirm authors's conclusion.
Response 1: Acetate_COM peaks were missing because of too intense normalization and smoothing that was done for the spectra to get them adjusted on the same graph. The peaks are present in the original spectra. To present the cellulose esters spectra more clearly, two separate graphs were made without normalization and less smoothing. Peaks are there with lower intensity than other synthesized esters.
All corrections in the given PDF file were corrected in manuscript.
Round 2
Reviewer 3 Report
The authors have revised the manuscript and addressed all issues proposed in my comments. I suggest that this work can be accepted now.
Minor English editing is required.